# T1w/T2w Ratio and Cognition in 9-to-11-Year-Old Children

**DOI:** 10.3390/brainsci12050599

**Published:** 2022-05-04

**Authors:** Lara Langensee, Theodor Rumetshofer, Hamid Behjat, Mikael Novén, Ping Li, Johan Mårtensson

**Affiliations:** 1Faculty of Medicine, Department of Clinical Sciences Lund, Logopedics, Phoniatrics and Audiology, Lund University, 22100 Lund, Sweden; theodor.rumetshofer@med.lu.se (T.R.); johan.martensson@med.lu.se (J.M.); 2Faculty of Engineering, Department of Biomedical Engineering, Lund University, 22100 Lund, Sweden; hamid.behjat@bme.lth.se; 3Faculty of Science, Department of Nutrition, Exercise and Sports, University of Copenhagen, 2200 Copenhagen, Denmark; noven@nexs.ku.dk; 4Faculty of Humanities, Department of Chinese and Bilingual Studies, The Hong Kong Polytechnic University, Hung Hom, Kowloon, Hong Kong, China; ping2.li@polyu.edu.hk

**Keywords:** T1w/T2w ratio, structural MRI, intracortical myelin, cognitive abilities, neurocognition

## Abstract

Childhood is a period of extensive cortical and neural development. Among other things, axons in the brain gradually become more myelinated, promoting the propagation of electrical signals between different parts of the brain, which in turn may facilitate skill development. Myelin is difficult to assess in vivo, and measurement techniques are only just beginning to make their way into standard imaging protocols in human cognitive neuroscience. An approach that has been proposed as an indirect measure of cortical myelin is the T1w/T2w ratio, a contrast that is based on the intensities of two standard structural magnetic resonance images. Although not initially intended as such, researchers have recently started to use the T1w/T2w contrast for between-subject comparisons of cortical data with various behavioral and cognitive indices. As a complement to these earlier findings, we computed individual cortical T1w/T2w maps using data from the Adolescent Brain Cognitive Development study (N = 960; 449 females; aged 8.9 to 11.0 years) and related the T1w/T2w maps to indices of cognitive ability; in contrast to previous work, we did not find significant relationships between T1w/T2w values and cognitive performance after correcting for multiple testing. These findings reinforce existent skepticism about the applicability of T1w/T2w ratio for inter-individual comparisons.

## 1. Introduction

Extensive neurodevelopment during childhood goes hand in hand with the acquisition and refinement of a wide range of skills. One striking manifestation of this is the rapid gain of cognitive abilities during the first years of life. The cerebral cortex is seen as a hub of higher cognition [1], and the advent of non-invasive neuroimaging has provided researchers with ample opportunities to study how ongoing cortical changes during childhood map to emerging cognitive abilities. The specific patterns of development are complex and differ regionally, as well as depending on the underlying cellular processes. Generally speaking, however, children’s cortical development is characterized by an initial growth spurt during the first two years of life with increases in cortical volume, thickness and surface area, followed by protracted periods of gradual decrease interleaved with relative stability during later childhood and beyond [2,3].

The interplay of cortical and neurocognitive development has been studied relatively more in relation to different indices of brain macrostructure [4,5,6,7] but is less well understood with respect to microstructural properties. One of the candidate mechanisms that has been suggested to underlie cortical reshaping and cognitive development during childhood is axonal myelination [2,8]. After years at the relative periphery of the scientific focus, myelin and its role in learning and memory formation have lately gained increasing recognition in the field of human cognitive neuroscience. Myelin, a fat-rich substance produced by glial cells, coats the brain’s nerve fibers, promoting communication between spatially disjoint brain areas [9,10], an essential prerequisite for healthy brain function [11]. The formation of myelin in the brain is an important maturational process during childhood, and it appears to be tightly coupled to cognitive development [12,13,14]. Traditionally, myelin in humans has been researched mainly in the context of demyelinating diseases (most notably multiple sclerosis), but as of recently, the scientific community has started to explore its impact on brain function in a broader, non-clinical sense [9,15,16]. Despite the complexities of non-invasive myelin content measurements [9,17,18], the development and use of techniques to estimate myelin content in humans have started to gain momentum [16,19]—perhaps encouraged by the growing evidence on the crucial role of myelin in the biology of learning from animal and cell culture research [20,21,22]. Histological studies are optimal for accurately assessing myelin content, but beyond that, various magnetic resonance imaging-based (MRI) techniques to estimate myelin content in the brain in vivo have also been suggested [9,17,18,23].

As a case in point, Glasser and Van Essen [24] have proposed to divide a T1w by a T2w image [9] as a proxy measure of cortical myelin, capitalizing on the fact that myelin—the primary source of the gray-white-matter contrast in MRI [25]—is highly correlated with the T1w intensity while being inversely correlated with the T2w intensity [26,27]. Since both images are equally affected by the scanner’s receive bias field, the effects are reduced in the ratio image [24]. To facilitate the use of T1w/T2w contrasts at a group level, Ganzetti and colleagues [17] proposed a normalization scheme, which addresses random variations in signal intensities that can occur due to external factors, such as hardware, protocol, or participant parameters. Although the T1w/T2w contrast appears better suited for assessing myelin in cortical than subcortical regions [28,29,30], adaptations for subcortical myelin measurements have also been explored [31,32]. The T1w/T2w ratio correlates moderately with the diffusion MRI indices fractional anisotropy and axial diffusivity, suggesting that the measures are all sensitive to myelin to some degree. At the same time, the findings point toward each metric also reflecting additional tissue properties, such as fiber density and orientation [30]. T1w/T2w contrast imaging appears to be a better fit for cortical segmentation than for longitudinal comparisons of cohorts—though in both cases, ensuring appropriate measures to reduce residual bias fields is essential [33]. In a recent effort to examine the reproducibility and reliability of the T1w/T2w ratio, Nerland and colleagues [34] highlighted issues related to bias field correction and intensity normalization and provided guidelines for avoiding systematic distortions dependent on dataset characteristics and intended use of the measure. Their findings point toward problems with the test–retest reliability of the uncorrected T1w/T2w ratio, and as such, the authors recommend applying normalization procedures if the contrast is used for group comparisons or correlational analyses [34].

Research on the relationship between T1w/T2w ratio maps and behavioral data in humans is limited, but some reports exist in the literature that link the measure to various cognitive abilities across a wide range of ages [35,36,37,38,39]. The T1w/T2w contrast has also been applied within clinical and affective neuroscience [40,41,42,43,44,45,46]. Overall, these results are encouraging, in that they point to the T1w/T2w ratio’s potential to reveal relevant information about the relationship of intracortical microstructure and cognitive abilities and other indices of behavior, as well as the feasibility of studying larger cohorts with this technique. At the same time, however, the authors who proposed the T1w/T2w ratio themselves advise against using the measure for group comparisons or correlations with biological or behavioral variables [24,47], primarily because the ratio is based on raw image intensities that are inherently devoid of a standardized measurement unit, and thus, its interpretability across individuals is limited.

Considering these conflicting perspectives, the objective of the present study was to explore the T1w/T2 ratio and its relation to cognitive performance in a large sample that is more homogeneous in age than what has been studied previously. The cautious assumption underlying this work is that a large enough sample may, to a certain extent, help extenuate some of the drawbacks of making interindividual comparisons based on a unitless measure, particularly in light of various available normalization procedures that were proposed to facilitate between-individual analyses based on the T1w/T2w ratio [17,34]. Proceeding from the emerging literature on statistical associations between cortical T1w/T2w maps and indices of behavior, this study can be viewed as an attempt to probe the tentative knowledge we have from previous reports in a different cohort—particularly in light of recently renewed criticism toward using the measure in this way. Fragmented and partially conflicting experimental findings are a prevailing problem in cognitive neuroscience, and large-scale data initiatives can be one potential remedy for this (for an overview of issues regarding the replicability of structural brain-behavior associations, see, for example, Ref [48]). Using baseline structural MRI and cognitive performance data from the Adolescent Brain Cognitive Development (ABCD) study, we investigated how intracortical T1w/T2w ratio relates to cognitive abilities in 9-to-11-year-olds, while controlling for basic demographic variables that have previously been found to correlate with brain structure [4,7,49,50]. Since not much is currently known about the patterns of association between cognitive performance and T1w/T2w ratio in this age group, this study takes an exploratory approach, examining links between microstructure and cognition across the entire cortical surface.

The literature suggests that primary sensory cortices are myelinated earlier than higher-order cortical and limbic areas and the insular cortices [51,52,53,54]. In addition, when investigating a cross-sectional sample spanning almost eight decades, Grydeland et al. [52] showed that the age at peak myelination was bimodally distributed, with primary sensory areas reaching their highest point before puberty, while associative cortical areas did not reach theirs until after puberty. Based on this literature, we expected visual, auditory and motor cortices to be more heavily myelinated than higher-order processing areas.

## 2. Materials and Methods

*Participants.* The MRI and cognitive performance data used in the present analysis were collected as part of the baseline assessment for the ABCD study. This longitudinal, large-scale research project acquired a wealth of different types of data from nearly 12,000 US children to identify the internal and external factors that can affect an individual’s developmental trajectory [55]. The data are available to qualified researchers via a repository managed by the National Institute of Mental Health Data Archive (NDA; https://nda.nih.gov/abcd, accessed on 17 April 2022). The data collection was launched in 2017 and is planned to continue for a period of 10 years at 21 research institutions across the United States. The cohort, aged between 9 and 11 years at the first timepoint, reflects the ratio of genders and ethnic and socio-economic backgrounds present in the general population in the US in that age group. Most of the universities involved in the ABCD data collection rely on a central Institutional Review Board (located at UC San Diego) for ethical approval and review; the remaining research sites work together with local Institutional Review Boards [56]. For details on how the ABCD Research Consortium ensures standards of ethical research conduct, see Refs [56,57,58].

We preprocessed MRI data for a randomly chosen subset of ABCD participants (N = 1000). Only those who had both a T1w and T2w image at the baseline assessment, which had passed the ABCD quality control procedures (see below for details), as well as scores for all seven cognitive measures, qualified for inclusion in the final sample. Due to an increased risk of atypical language lateralization in left-handers [59], only right-handed participants were included. These criteria left us with a final sample of N = 960 for our analysis (511 males, 449 females). An overview of demographic information on the sample can be found in Table 1.

*Cognitive performance.* Cognitive performance was assessed by means of seven computerized tasks, all of which are part of the NIH Toolbox Cognition Battery. This toolbox is designed to capture different cognitive constructs, measuring both fluid and crystallized abilities; see Table 2 for an overview of the tasks and the cognitive processes they assess. More detailed information on each of the tasks can be found elsewhere [60,61]. Age-corrected standard scores were used for all analyses. All measures were administered on an iPad using computerized adaptive testing, and all children completed the tasks in English [61]. Visualizations of the performance distribution for each cognitive domain can be found in Appendix A.

*Neuroimaging*. Three-dimensional T1w (1 mm isotropic) inversion prepared RF-spoiled gradient echo and three-dimensional T2w (1 mm isotropic) variable flip angle fast spin echo structural images (in both cases using prospective motion correction when available [62]) were collected on 3T scanners from three different manufacturers (Siemens Prisma and Prisma Fit, GE MR 750, and Philips Achieva, dStream and Ingenia [62]). Detailed information about the acquisition protocol can be found in Ref [55]. All data in this study stem from the ABCD Curated Annual Release 3.0 (http://dx.doi.org/10.15154/1519007; accessed on 22 April 2022), which means that the T1w and T2w images have already undergone some basic preprocessing, such as correction procedures for gradient nonlinearity distortions and intensity inhomogeneities. For a detailed account of the processing steps performed on the curated releases, see Ref [62].

*MRI quality control*. Following in-house preprocessing, the data from the ABCD Curated Annual Releases undergo manual inspection by trained reviewers to ensure a consistent quality level [62]. Images are assessed for five types of artifacts and reconstructions inaccuracies: motion, intensity inhomogeneity, white matter underestimation, pial overestimation and magnetic susceptibility artifact. If any of these is deemed severe, the usage of the respective cortical surface reconstruction is advised against [62]. For this analysis, only recommended MRI data were included.

*MRI processing*. As part of the ABCD Curated Annual Release 3.0, the MRI data have already undergone some basic preprocessing. Details about the procedures can be found in Ref [62]. The T1w and T2w images were further processed with the three minimal structural preprocessing pipelines (v 4.2.0) from the Human Connectome Project (HCP), described in depth in Ref [63], which are implemented in FreeSurfer (Image Analysis Software, v 6.0.0) [64] and FSL (FMRIB Software Library, v6.0.4) [65]. The first of the three pipelines, the PreFreeSurfer pipeline, generates a native structural volume space for each participant, aligns the T1w and T2w images, performs bias field correction and registers the individual’s native structural volume space into MNI space [63]. In the next step, the FreeSurfer pipeline, volumes are segmented into predefined structures, white and pial surfaces are reconstructed, and images are registered to FreeSurfer’s surface atlas *fsaverage* [63]. The last pipeline, the PostFreeSurfer pipeline, creates volume (NifTI) and surface (GifTI) files, performs surface registration (to the Conte69 template), downsamples the output and generates final brain masks and myelin maps based on the T1w/T2w contrast proposed by Glasser and Van Essen [24], including a few subsequent adjustments [63,66] that help avoid surface reconstruction errors. Voxels whose T1w/T2w values deviated more than one standard deviation from all T1w/T2w values inside the cortical ribbon were excluded (Glasser and Van Essen, 2011). PostFreeSurfer outputs both smoothed (4 mm Gaussian filter) and unsmoothed myelin maps. The results reported below are based on unsmoothed maps. All analyses were also performed on smoothed maps—the output of which did not differ significantly (i.e., nearly perfect spatial overlap between smoothed and unsmoothed clusters and only minimal variations in *p*-values) from the results presented below.

*Statistics*. Separate general linear models (GLM), as implemented in FSL PALM (Permutation Analysis of Linear Models; [67]), were fit to test for possible associations (both positive and negative) between vertex-wise T1w/T2w ratio and the performance on each of the seven cognitive tests. In all models, 22 scanner sites were dummy coded and included as covariates of no interest (In the early stages, data for the ABCD study was collected at 22 sites. One of these is no longer active, which is why most ABCD documentation refers to 21 rather than 22 sites). To account for kinship between some of the subjects, exchangeability blocks reflecting family structure were added to the models [67] to prevent data being shuffled between subjects from different families. In addition, we modeled the relationship between cortical T1w/T2w ratio and age, sex and socioeconomic status (SES) within a separate analysis, as existent literature suggests these variables to be related to brain structure [4,7,49,50]. SES was operationalized as a binary variable specifying the highest level of parental educational attainment, with the two possible values ‘Bachelor’s degree or higher’ and ‘Less than bachelor’s degree’, resulting in two roughly equally sized groups. Age (in months) was added to the model as a continuous variable, while sex and SES were dummy coded. Due to some missing data for the SES variable, the sample for all demographic analyses is slightly smaller than the original sample (N = 953).

To assess statistical significance, 10,000 permutations and family-wise error (FWE) correction—across vertices, hemispheres and contrasts (positive and negative correlations between T1w/T2w ratio and the respective cognitive test)—with threshold-free cluster enhancement (TFCE) [68] were used, with a significance threshold of *p* < 0.05, after permutation-based correction for multiple testing.

## 3. Results

The individual cortical T1w/T2w ratio maps confirm earlier findings [51,52,53], in that they show a clear tendency for myelin peaks in primary sensory areas (visual, somatosensory and motor cortices) as compared with higher-order processing regions, such as the pre-frontal cortex. Figure 1 depicts the sample’s mean T1w/T2w ratio map. At the same time, they clearly illustrate a considerable amount of individual variation in terms of the unique patterns and time-dependent progression of myelination—see Appendix A for three individual myelin maps from age-matched participants.

None of the demographic regressors (age, sex and SES) was associated with regional cortical T1w/T2w ratio at corrected alpha = 0.05. Unthresholded t-statistical maps depicting the individual contrasts are available as Appendix A (Appendix A).

No significant associations between cortical T1w/T2w ratio and any of the cognitive measures emerged in the FWE-corrected analyses at alpha = 0.05. Re-examining the data without applying multiple testing correction resulted in significant negative correlations between the T1w/T2w values and working memory (as indicated by the List Sorting Working Memory Test), as well as language (as assessed by the Picture Vocabulary Test) and positive associations between T1w/T2w ratio and processing speed (as indicated by the Pattern Comparison Processing Speed Test) at an alpha threshold of 0.05. At a stricter alpha level of 0.01, a few peaks remain for regional associations between T1w/T2w values and working memory, respectively, processing speed. When it comes to language performance, thresholding the maps at alpha = 0.01 eliminates the correlations with T1w/T2w, except for a few, likely spurious, vertices in the inferior portion of the medial surface of the left hemisphere (since the -log10 of the peak vertices is only marginally higher than −log10(*p*) ≥ 2). For illustrative purposes, t-statistical maps are given in Figure 2, Figure 3 and Figure 4 showing the areas in which T1w/T2w values correlated significantly with cognitive performance at uncorrected alpha = 0.05, respectively, alpha = 0.01; *t*-statistical maps are projected onto the average inflated cortical surface of the sample. Unthresholded, FWE-corrected statistical maps for the remaining cognitive measures are available as Appendix A.

Beyond what is reported here, we also tested for positive and negative correlations between vertex-wise, cortical T1w/T2w ratio and cognitive performance with age, sex and SES included in the model as covariates of no interest. The findings from these analyses correspond well with that we found based on the models that tested for the main effect of cognitive performance while only regressing out the effects of the scanner site. Unthresholded, FWE-corrected t-statistical maps from these analyses are available in the Appendix A (Appendix A).

## 4. Discussion

The relationship between myelin and behavior has lately been drawing great interest in human cognitive neuroscience, and a small number of recent publications have reported cortical T1w/T2w values to be linked to different indices of cognition. Following a similar approach, we did not observe any such brain-behavior relationships in the subset of the ABCD data that we analyzed. Despite working with a sizeable sample and a very limited age range, our analysis did not yield any significant associations between cortical T1w/T2w values and any of the assessed cognitive domains when applying FWE-corrections to adjust for multiple testing. The question that arises from this is whether the explanatory power of interindividual differences in the cortical microstructure alone is insufficient to account for performance differences in specific cognitive abilities in this population or whether the lack of findings in the present analysis may instead rather be caused by an inadequately chosen method. Indices of white matter microstructure have previously been found to explain cognitive performance differences in typically developing children [69,70,71]. In view of this, it does not seem far fetched to expect a similar pattern with respect to cortical microstructure. What should be kept in mind in this regard is that Glasser and colleagues [24,47] presented the T1w/T2w contrast primarily as a tool for cortical parcellation and not as a suitable measure for inter-individual comparisons or correlational brain-behavior analyses. The obvious appeal of the technique lies in its simplicity: T1w and T2w images are routinely acquired in most MRI examinations. The acquisition times are reasonable, even for higher resolutions, and the computation of the ratio image is relatively straightforward compared to other indices of myelin that typically rely on complex modeling procedures [9]. However, bearing in mind that the T1w/T2w ratio is not a quantitative measure—meaning that the values it yields are unitless—it becomes apparent why interpreting its meaning across participants can be problematic. The present study is in line with this to the effect that it does not provide any evidence for associations between cortical T1w/T2w ratio and cognitive performance.

Our finding conflicts with earlier studies that point toward systematic links between cortical T1w/T2w ratio and different indices of cognition across a wide range of ages [35,36]. T1w/T2w ratio has, for instance, been associated with performance stability, especially in older participants—perhaps indicating that age-related cortical demyelination may give rise to larger variability in individual performance [35]. Similarly, T1w/T2w ratio in white matter and subcortical structures correlates positively with intelligence, language and visuo-motor skills in children [37]. On the other hand, better overall cognitive ability in children and young adults has been related to lower intracortical myelin in frontal regions [36]. Inverse correlations were also observed between local T1w/T2w ratio and more specific aspects of cognition, ranging from attention, inhibition and language to working memory [36]. At the same time, a recent study [39], based on the same dataset as the present work, only found one statistically significant association between T1w/T2w ratio and a composite score of cognitive performance in an atlas-based analysis. One conceivable cause behind this discrepancy is the different age groups that were investigated in the studies. While the present work focused on a relatively narrow age bracket (min = 8.9 years, max = 11.0 years), earlier research has applied the T1w/T2w ratio to samples covering much wider age ranges (3–21 years in Ref [36]; 8–83 years in Ref [35]). It should be added that some of Norbom and colleagues’ [36] findings appear very intuitive, while others seem less plausible. More specifically, the linear across-cortex increase in myelin between early childhood and beginning of adulthood they observed, aligns well with what one would expect based on the existent literature. The inverse association between myelination and several cognitive abilities [36], on the other hand, is a more puzzling result when viewing the T1w/T2w ratio as a proxy for cortical myelin content. As it happens, two of the three cognitive measures that yielded significant results in our sample when correction procedures were omitted (working memory and receptive language abilities) correlated negatively with cortical T1w/T2w values, corroborating Norbom and colleagues’ [36] findings. However, this result should be treated with caution, given the uncorrected, whole-brain voxel-wise analyses that were used. Relating this outcome to earlier studies that also discovered unexpected links between brain structure and cognition [35,72], Norbom et al. [36] conjectured that this observation may be explained by the inhibitory effect of myelin on axonal sprouting and synaptogenesis [73], since excessive myelination might impede neuronal plasticity to an extent that becomes detrimental during development. The current literature on this topic is scarce, which thwarts a reliable answer for the time being. Additional studies, especially ones that employ alternative techniques, ideally quantitative in nature, such as R1 [74], to assess myelin, will be needed to shed light on the relationship between cognition and myelin in children.

The early years of life are characterized by extensive myelination [10], and the current data indicate that this process unfolds differently in every individual. Even so, it may be the case that the T1w/T2w contrast can perform well for brain-behavior correlations when a wide variety of ages—and thus stages of both cognitive and cortical development—are taken into account, but less so, when a very specific age group is considered. Similarly, it is possible that the T1w/T2w ratio is more suitable for contrasting clinical populations with healthy individuals, where comparably larger differences in local myelin content can be expected [75,76,77]. Nevertheless, even within this context, caution should be exercised, as empirical findings do not always align well with the assumption that T1w/T2w ratio can be interpreted as a proxy for intracortical myelin; for example, Alzheimer’s patients have been found to have higher T1w/T2w values than cognitively normal controls [77].

Aside from the evident shortcoming of applying a technique outside of its intended scope, there are a few additional factors that may have reduced the accuracy of the T1w/T2w ratio as an indicator of cortical myelin content in this study. Given that the structural images that were used in the present analysis have a 1 mm isotropic resolution, the T1w/T2w contrast may not be sensitive enough to detect relatively small-scale differences in myelination, considering that the average thickness of the human cortex lies somewhere around 2.5 mm, with relatively large regional variations [78]. Partial volume effects give reason for concern in this context. Here, image intensities can be distorted in areas where gray and white matter and cerebrospinal fluid are found near one another, which is especially the case for the cortical ribbon [11]. However, both Norbom et al. [36] and Grydeland et al. [35] had comparable resolutions to ours, so it does not seem plausible to assume that the null findings in this sample can be attributed to partial volume effects alone. That said, an accurate reconstruction of the white and pial surfaces is essential for producing truthful myelin maps [24], and typically, surface reconstruction benefits from high resolution, and it could thus be speculated that the results of the present study may have looked different if images with submillimeter resolution had been available.

There is no consensus on how accurately the T1w/T2w contrast reflects myelin content. Results from prior work have been promising in terms of the ratio’s ability to describe the development of myelination over time [11], but comparisons with other measures of myelin have led to inconsistent results [28,29,30,79]. Presumably, this mismatch originates at least partially from the fact that all these measures reflect other tissue properties than myelin to varying extents. The lack of biological specificity, however, is not an issue that is unique to the T1w/T2w ratio, but rather one that affects all MRI-based measures to some degree.

Lastly, it could be the case that the relationship between myelination and behavioral indices is more complicated than what is captured by a linear model, especially in a population that is undergoing a period of extensive neural reshaping and remodeling. What we know so far seems to suggest that myelination boosts skill development. So, to put it crudely, one would expect that more myelin equals more ability. At the same time, previous research indicates that myelination is not a linear process throughout the life span but rather that it proceeds in waves and that different regions follow different timelines [15,52,54]. Since the current sample is constrained with respect to the included ages, one could argue that such a snapshot of development should lend itself especially well to revealing correlations with behavior. One factor behind the current result might be that the T1w/T2w ratio is simply better suited for implementations with diverse populations, for example across a wider range of ages or to distinguish between patients and healthy volunteers, whereas it may not be quite fine grained enough to characterize a healthy, more homogenous group, such as the young ABCD participants at the baseline assessment. Given that earlier studies have suggested links between cognition and cortical morphology [5,80], it seems probable that regional associations between cognitive performance and cortical myelin do exist. To what extent T1w/T2w ratio mapping is an appropriate measure to uncover these, however, is doubtful in view of the current findings. Going forward, this work should be complemented by similar studies using B1+ corrected T1w/T2w ratio maps [47] as well as alternative myelin measures to eventually determine whether the null findings presented here reflect a biological reality or whether they are merely the result of applying a technique outside its intended scope.

## 5. Conclusions

Aside from not yielding evidence for systematic links between cortical T1w/T2w ratio and cognition in 9–11-year-old children, the current findings suggest that skepticism is warranted when it comes to incorporating the uncorrected T1w/T2w ratio into interindividual comparisons of performance variables. First and foremost, the authors who proposed this measure did not intend it to be used for between-subject statistical analyses [24,47]. At the same time, however, various propositions have been made since Glasser and colleagues [24] first presented their method about how it could be modified to enhance its stability across sites and individuals and, thereby, to expand its field of application [11,17,34,47]. These adaptations will require gradual validation—for example, by comparing them with other measures of myelin—paving the way for reliable and high-specificity in vivo assessments of myelin content. Such developments will be crucial for scientists to ultimately gain a better understanding of concurrent cortical and cognitive development.

## Figures and Tables

**Figure 1 brainsci-12-00599-f001:**
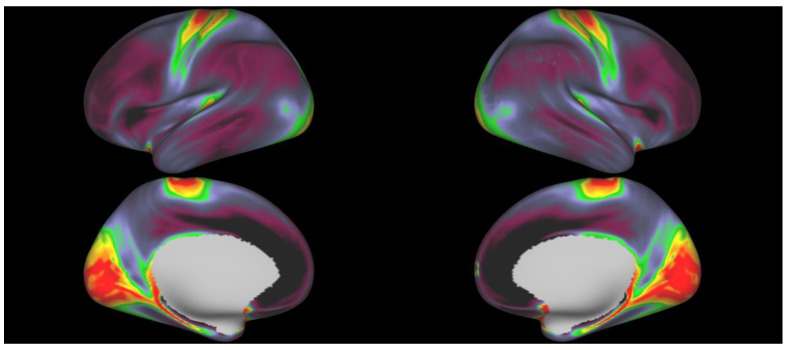
Sample’s (N = 960) average T1w/T2w ratio map, projected onto its average inflated surface. Warm colors are higher, cool areas lower in myelin.

**Figure 2 brainsci-12-00599-f002:**
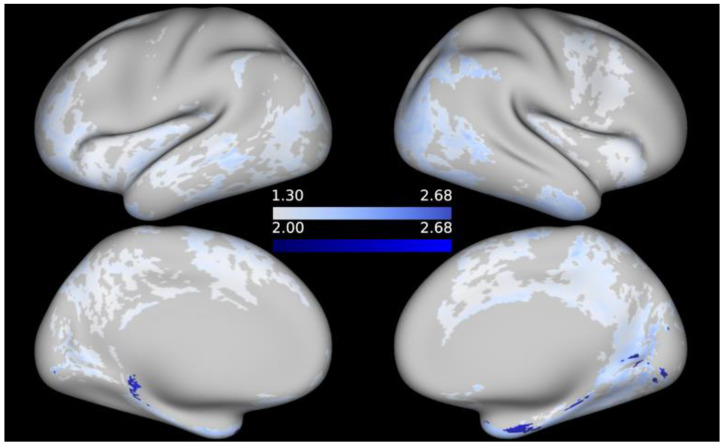
*t*-statistical map representing areas in which working memory was negatively associated with T1w/T2w contrast. Thresholded at −log10(*p*) ≥ 1.3 (pale blue regions), respectively −log10(*p*) ≥ 2 (opaque); *p*-values are uncorrected.

**Figure 3 brainsci-12-00599-f003:**
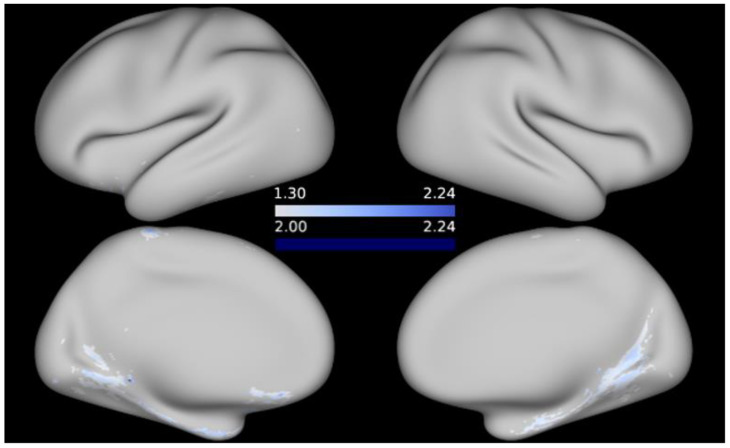
*t*-statistical map representing areas in which language ability was negatively associated with T1w/T2w contrast. Thresholded at −log10(*p*) ≥ 1.3 (pale blue regions); *p*-values are uncorrected.

**Figure 4 brainsci-12-00599-f004:**
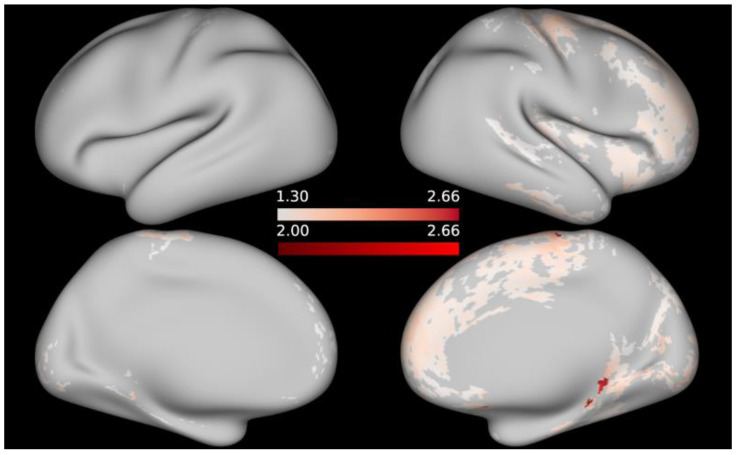
*t*-statistical map representing areas in which processing speed was positively associated with T1w/T2w contrast. Thresholded at −log10(*p*) ≥ 1.3 (pale orange regions), respectively −log10(*p*) ≥ 2 (opaque); *p*-values are uncorrected.

**Table 1 brainsci-12-00599-t001:** Basic demographic information ^1^.

Demographic Variable	Females N = 449	Males N = 511
Age (in years)	M = 9.97 (SD = 0.61)	M = 10.04 (SD = 0.62)
Race	White = 347	White = 417
	African American = 79	African American = 66
	Native American = 15	Native American = 7
	Asian = 25	Asian = 29
	Other = 37	Other = 33
Household income (in the past 12 months)	USD 200 k and greater = 46	USD 200 k and greater = 53
	USD 100 k–199 k = 141	USD 100 k–199 k = 158
	USD 75 k–99,999 = 73	USD 75 k–99,999 = 72
	USD 50 k–74,999 = 43	USD 50 k–74,999 = 69
	USD 35 k–49,999 = 47	USD 35 k–49,999 = 43
	USD 25 k–34,999 = 21	USD 25 k–34,999 = 23
	USD 16 k–24,999 = 19	USD 16 k–24,999 = 30
	USD 12 k–15,999 = 8	USD 12 k–15,999 = 8
	USD 5 k–11,999 = 17	USD 5 k–11,999 = 10
	Less than USD 5000 = 5	Less than USD 5000 = 7
Parental Education ^2^	Bachelor’s degree or higher = 264 Some form of post-high school education = 129	Bachelor’s degree or higher = 287 Some form of post-high school education = 156
	High school degree = 32	High school degree = 39
	No high school degree = 23	No high school degree = 28

^1^ Numbers do not always add up to N = 960 due to missing data and multiple responses. ^2^ Referring to the parent that filled out the questionnaire.

**Table 2 brainsci-12-00599-t002:** Overview of the cognitive measures from the NIH toolbox.

Task	Cognitive Domains
Oral Reading Recognition Test	Language Reading Decoding
Picture Vocabulary Test	Language Receptive vocabulary
Flanker Inhibitory Control and Attention Test	Executive functioning Attention Inhibitory control
Dimensional Change Card Sort Test	Executive functioning Cognitive flexibility
Picture Sequence Memory Test	Visuospatial sequencing
	Episodic memory
List Sorting Working Memory Test	Working memory
	Information processing
Pattern Comparison Processing Speed Test	Processing speed

## Data Availability

Data used in the preparation of this article (http://dx.doi.org/10.15154/1519007; accessed on 2 May 2022) were obtained from the Adolescent Brain Cognitive Development (ABCD) Study (https://abcdstudy.org, accessed on 15 March 2022), held in the NIMH Data Archive (NDA). This is a multisite, longitudinal study designed to recruit more than 10,000 children aged 9–10 and follow them over 10 years into early adulthood. The ABCD Study^®^ is supported by the National Institutes of Health and additional federal partners under award numbers U01DA041048, U01DA050989, U01DA051016, U01DA041022, U01DA051018, U01DA051037, U01DA050987, U01DA041174, U01DA041106, U01DA041117, U01DA041028, U01DA041134, U01DA050988, U01DA051039, U01DA041156, U01DA041025, U01DA041120, U01DA051038, U01DA041148, U01DA041093, U01DA041089, U24DA041123, U24DA041147. A full list of supporters is available at https://abcdstudy.org/federal-partners.html (accessed on 15 March 2022). A listing of participating sites and a complete listing of the study investigators can be found at https://abcdstudy.org/consortium_members/ (accessed on 15 March 2022). ABCD consortium investigators designed and implemented the study and/or provided data but did not necessarily participate in the analysis or writing of this report. This manuscript reflects the views of the authors and may not reflect the opinions or views of the NIH or ABCD consortium investigators.

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
