# Peer review of "T1w/T2w Ratio and Cognition in 9-to-11-Year-Old Children"

_brainsci, 2022, doi:10.3390/brainsci12050599_

Round 1

Reviewer 1 Report

The manuscript reports on (lack of) associations between MRI-derived myelin-content maps (from T1w/T2w ratios) in a large sample of children from a publicly available data from the ABCD study. the study is well motivated and executed and the manuscript is very clearly written. I highly appreciate the intellectual honesty of the authors in the reporting of null results of their analyses and I think the paper constitutes an important contribution to the field. However I think that for the results to be more impactful the paper could profit from several adjustments.

In the revision please address the following issues:

(1) The authors do not control for any confounding variables that in previous pediatric neuroimaging studies have been shown to be important for brain structure. I would strongly encourage the authors to rerun their analyses controlling for factors such as gender (cf. Schmithorst et al. 2008), age (cf. Mills et al. 2016) and SES (cf. Brito & Noble 2014). Note that in a developmental sample such as the one included in the present study, even a narrow age range might have an effect on the results. It would be furthermore insightful to see if the above mentioned variables are in any way related to voxel-wise myelin content.

(2) The authors present an interesting result with regards to the spatial distribution of myelination across different cortical areas (Figure 4). It would be really interesting to explore this result further and see whether it is generalizable to the whole sample and whether it is robust against the effect of the potential confounders mentioned above. I would therefore urge the authors to please report whole brain mean myelination values for the whole group in which the effect of the demographic confounders is controlled for. 

Furthermore, since this analysis is a more general one (i.e., it investigates a general pattern of myelination across the brain), I would start your results reporting with it, before proceeding to the more detailed analysis involving cognitive tests performance. (The order of referring to these analyses should then also be changed in the introduction.)

(3) Space permitting, if the analysis involving cognitive measures incorporating the demographic confounding variables still fails to produce any (corrected) significant results, I would encourage the authors to accompany their frequentist statistical inferences by an analysis involving Bayesian inference. This way, the authors could interpret their null results with more certainty. Existing tools for model selection of neuroimaging data include for example the MACS toolbox (Soch et al. 2018, see also https://github.com/JoramSoch/MACS) that has so far been used for fMRI model selection, but that could potentially equally lend itself for comparison of group-level voxel-wise myelin maps in nifti format. Comparing GLMs including  cognitive measures of interest against baseline GLMs with confounding variables only with e.g., log Bayes factor maps (produced by subtracting two MACS-generated voxel-wise cross-validated log model evidence [cvLME] maps from each other) would allow the authors to interpret the lack of the significant results with more precision and assess whether the non-significant p values in the frequentist analyses are a result of no relationship between the investigated variables or lack of sensitivity in the data.

References cited:

Brito, N. H. and Noble, K. G. (2014) ‘Socioeconomic status and structural brain development’, Frontiers in Neuroscience. Frontiers Research Foundation, 8(SEP), p. 276. doi: 10.3389/FNINS.2014.00276/BIBTEX.

Mills, K. L. et al. (2016) ‘Structural brain development between childhood and adulthood: Convergence across four longitudinal samples’, Neuroimage. Elsevier, 141, p. 273. doi: 10.1016/J.NEUROIMAGE.2016.07.044.

Soch, J. and Allefeld, C. (2018) ‘MACS – a new SPM toolbox for model assessment, comparison and selection’, Journal of Neuroscience Methods. Elsevier, 306(May), pp. 19–31. doi: 10.1016/j.jneumeth.2018.05.017.

Schmithorst, V. J., Holland, S. K. and Dardzinski, B. J. (2008) ‘Developmental differences in white matter architecture between boys and girls’, Human Brain Mapping. Wiley-Blackwell, 29(6), p. 696. doi: 10.1002/HBM.20431.

Reviewer 2 Report

The current study assessed relations between cortical microstructure as measured by vertex-wise T1w/T2w-ratio and several specific cognitive abilities in 940 typically developing children aged 9-10 years. They found no significant association between T1w/T2w-ratio and youth cognition.

The study, to my knowledge, has the largest sample to date, assessing T1w/T2w ratio and specific cognitive abilities in childhood. The narrow age range is also a major strength, as previous studies using wide age ranges struggle with disentangling age and cognitive ability as the two show strong correlations in youth. I also commend the authors for wanting to publish a null finding, as this is critical for combatting the current replication crisis. Unfortunately, I have several major issues with the paper, including not always following standard scientific writing structure, not providing enough transparent imaging and statistical methods information, and most importantly appearing to change the theme of the paper post hoc due to the null finding. I have detailed my comments below, and sincerely hope they can be of some help, as publishing null findings are critical in current neuroscience.

Major general comments:

- The abstract is mostly focused on intracortical myelin and the specific MRI metric. Similarly, the introduction is highly unbalanced with little general information about youth cognition and brain development but a strong neurobiology and T1w/T2w ratio focus. I would suggest dedicating more space to neurocognitive development in youth, and less on detailing the imaging metric. I would also tone down the focus on intracortical myelination, as MRI in general and T1w/T2w ratio specifically, are proxy measures. Also, describe your final sample (n=940), and not the general ABCD sample within these sections, and save general ABCD info for the “Participants” section.

-The methods section is too short, general and is lacking several standard details.

-It currently comes across, because of the null finding, that the paper moves away from its initial goal, and toward a critical methods paper of the T1w/T2w ratio, where the null finding is somehow used as an argument for the MRI measure both being a poor myelin proxy and unsuitable for assessments across subjects. If the researchers from the start wished to critically assess T1w/T2w ratio, it is unclear why they did not instead compare the metric either to other standard morphometric or signal intensity measures, or to the more superior myelin proxies of quantitative relaxometry. It is also unclear why the authors assess links to cognition as there is little focus on this within the paper. To be clear, I completely agree that the T1w/T2w ratio is a highly debated imaging metric with numerous challenges. I would suggest clarifying the theme of the paper as either an assessment of cortical microstructure and cognition with the strength of a limited age range, or as a T1w/T2w ratio methods paper, but then adding appropriate tests beyond failing to find an association with youth cognition.

Minor comments:

Abstract: Please provide total sample size and age and sex distribution of current sample

Introduction:

-Rephrase non-scientific wording such as: “the formation of myelin in the brain is in full swing». I believe there are a couple of other similar sentences throughout.

-The description of previous T1w/T2w ratio studies are too detailed, I would suggest moving such details to discussion, in comparison to current null-finding.

-I would rephrase “dividing a T1w and a T2w image” to “dividing a T1w by a T2w image”.

-I would rephrase so that image division “reduces” scanner’s receive bias field, and not “cancels out”.

-Currently there are no hypotheses related to any of the specific cognitive abilities, and a hypothesis regarding region specificity which I believe was not statistically addressed within the paper (whether early sensorimotor regions have statistically higher T1w/T2w ratio than higher order regions).

Participants:

-Please provide detailed age, sex and other demographics information of your final sample with for instance mean and SD values.

-Please provide more detail on how the 40 subjects were excluded.

-Why were only 1000 individuals chosen instead of the full abcd sample?

Other methods:

- “Cognitive performance” now comes in the middle of several MRI paragraphs.  

-How many scanners, and how many scan sites were included in your study? As a subset was chosen, were there attempts to choose subjects from a limited number of scanners?

-Was genetic ancestry handled somehow within the current study?

-Make clear that data was provided open access from ABCD and not “collected” by the researchers.

-What smoothing was performed on the T1w/T2w ratio maps?

-First line within “statistics” section regarding outlier voxels should be moved to MRI processing as it is a part of “PostFreesurfer”.

-Was age and sex added as co-variates in PALM? Please provide model details, and details on the block restrictions added. I believe block restriction also can be added based on scanner.

-Plots of cognitive ability data could be a good idea.

-What was the relations between cognition and age and sex in the current sample?

Results:

-Change “voxels” wording to “vertices”.

-I would suggest moving the individual T1w/T2w ratio maps to supplement.

Round 2

Reviewer 2 Report

The authors have greatly improved the original manuscript and replied to all my comments in detail. I still have some very minor comments that I hope the authors will consider, and please do not hesitate to correct me if I have misunderstood something.

- I would suggest consequently reporting the age of the subjects in years. Currently there are reports of age in months within the abstract and discussion.

-Introduction: I am not sure what is meant by-, and I would be careful with the term “higher cognitive abilities (during the first years of life)” within the introduction as it can be confused with “higher order cognitive functioning” such as executive functioning which I believe generally show advances in late childhood and adolescence.

-It is convention to use some form of smoothing when assessing surface based measures, including T1w/T2w ratio in order to: normalize distributions, minimize registration and anatomical misalignment across subjects and increase SNR and statistical power. I would strongly suggest using smoothed maps, for instance the ones outputted from HCP-pipeline which I believe is smoothed with a sigma kernel of 4. If not, state clearly within the methods section that un-smoothed maps were used for analyses.

-As stated in my original review letter the authors state an expectation, based on previous research, of sensory-motor regions showing higher T1w/T2w-ratio than frontal regions. I believe this hypothesis is not statistically tested but instead there appear to be descriptions from visual inspection of individual T1w/T2w ratio maps. If no tests are performed, I would suggest toning down the formal expectation (hypothesis) or for instance presenting a mean T1w/T2w ratio map coupled with the description already present in the results section.

-As I believe the age, sex and SES supplemental analyses now comes as a small surprise within the methods section, I would suggest adding a line within the introduction about also performing such complementary analyses.  

-I would state the number of scan sites used as a dummy-variable within PALM.

-I believe future T1w/T2w ratio studies will have to perform B1+ correction. I would add a line within the limitations section about not performing such correction based on this recent pre-print by Glasser (https://www.biorxiv.org/content/10.1101/2021.08.08.455570v4).
